# The Effect of Electric Aging on Vinylidene Fluoride Copolymers for Ferroelectric Memory

**DOI:** 10.3390/nano14121002

**Published:** 2024-06-09

**Authors:** Valentin V. Kochervinskii, Evgeniya L. Buryanskaya, Aleksey S. Osipkov, Mstislav O. Makeev, Dmitry A. Kiselev, Margarita A. Gradova, Oleg V. Gradov, Boris V. Lokshin, Alexandr A. Korlyukov

**Affiliations:** 1Laboratory of Technologies of Polymer Ferroelectrics, Bauman Moscow State Technical University, Moscow 141005, Russia; kochval@mail.ru (V.V.K.); m.makeev@bmstu.ru (M.O.M.); 2Laboratory of Physics of Oxide Ferroelectrics, Department of Materials Science of Semiconductors and Dielectrics, National University of Science and Technology MISIS, Moscow 119049, Russia; dm.kiselev@gmail.com; 3N.N. Semenov Federal Research Center for Chemical Physics (Russian Academy of Sciences), Moscow 119334, Russia; m.a.gradova@gmail.com (M.A.G.); o.v.gradov@gmail.com (O.V.G.); 4A.N. Nesmeyanov Institute of Organoelement Compounds RAS, Moscow 119334, Russia; bloksh@ineos.ac.ru (B.V.L.); alex@xrlab.ineos.ac.ru (A.A.K.)

**Keywords:** polymers, ferroelectric materials, flexible electronics, nanoelectronics, polarization, sensing technology, microstructure

## Abstract

Copolymers based on vinylidene fluoride are potential materials for ferroelectric memory elements. The trend in studies showing that a decrease in the degree of crystallinity can lead to an unexpected increase in the electric breakdown field is noted. An analysis of the literature data reveals that in fluorine-containing ferroelectric polymers, when using a bipolar triangular field, the hysteresis loop has an unclosed shape, with each subsequent loop being accompanied by a decrease in the dielectric response. In this work, the effect of the structure of self-polarized films of copolymers of vinylidene fluoride with tetrafluoroethylene and hexafluoropropylene on breakdown processes was studied. The structure of the polymer films was monitored using infrared spectroscopy (IR) and X-ray diffraction. Kelvin probe force microscopy (KPFM) was applied to characterize the local electrical properties of the polymers. For the films of the first copolymer, which crystallize in the polar β-phase, asymmetry in the dielectric response was observed at fields greater than the coercive field. For the films of the copolymers of vinylidene fluoride with hexafluoropropylene, which crystallize predominantly in the nonpolar α-phase, polarization switching processes have also been observed, but at lower electric fields. The noted phenomena will help to identify the influence of the structure of ferroelectric polymers on their electrical properties.

## 1. Introduction

The relatively recently discovered class of materials known as polymeric ferroelectric materials have several unique properties that make them a competitive alternative to inorganic materials in certain technical applications [1,2,3,4]. One of the possible and promising applications is the use of such materials to create a new class of memory elements based on ferroelectrics to replace ferromagnetic materials with them [5,6,7]. One of the requirements for such materials is the ability to withstand many switching operations. In this regard, work is still in progress to clarify the mechanisms of polarization switching in both inorganic [5,6,7,8,9,10,11,12,13] and polymeric [14,15,16,17,18,19,20] ferroelectrics.

In the former, it was noted that the parameters of memory elements are determined by a few material characteristics. For example, according to the results of [5,6], the role of spatial charge is important in this case. On the other hand, the same authors later note that, for perovskites, the processes of oxygen vacancy ordering play an important role in this case [8]. Since a Schottky barrier is formed at the metal–ferroelectric interface, electrode contacts must also be considered in such devices [13].

Materials for polymer-based memory elements currently include hybrid systems as well. They are discussed in more detail in recent works [3], and, therefore, they will not be discussed here. Let us limit ourselves to some results where interesting data were obtained mainly on polyvinylidene fluoride PVDF or on its copolymer with trifluoroethylene (TrFE). For example, an original procedure for the epitaxial preparation of highly oriented films of the noted copolymer was proposed in [14]. The substrate was a specially prepared textured polytetrafluoroethylene PTFE film, where the lattice parameters were similar to those of the P(VDF–TrFE) copolymer. Since the PTFE substrate has stable structure parameters up to 350 °C, the copolymer film formed with it also has increased thermal stability. Texture parameters were preserved up to 140 °C, which is significantly higher than the Curie point of the isotropic film.

It is of interest to examine the structural differences and stability parameters of polarization during multicycle switching in PVDF films and PVDF copolymers with trifluoroethylene of composition 72-28 [1,2]. It is known that copolymer films almost always crystallize immediately in the polar β-phase under such conditions, while PVDF can be formed with various polymorphic modifications [1,2]. To ensure the identity of the phase composition, PVDF crystallization was carried out in the presence of an agent providing the formation of a polar modification [1,2]. IR spectroscopy data show that both films exhibit absorption bands characteristic of the planar zigzag conformation. The X-ray data qualitatively support this conclusion, although, for the copolymer, the main reflex, 110,200, appears to be shifted to lower angles. This means that the lattice appears to be less densely packed than for PVDF copolymer due to the presence of TFE–CF_3_ groups in the copolymer chain, which exhibit steric obstacles when packing into the lattice [1,2,3]. As a result, the coercive field in the copolymer is lower and the depolarization process (the Curie point) occurs at lower values when the temperature is changed. The noted changes in the studied film structure ultimately have a beneficial effect on the stability of polarization during multi-cycle switching: PVDF films have a slightly higher window value. It is extremely important to dwell on the conclusion that authors draw at the end of their work. PVDF films have lower crystallinity, which is accompanied by an increase in their breakdown fields. Higher crystallinity in copolymer films is accompanied by the formation of defects in the form of microvoids, which are in the gaps between neighboring lamellar crystals. This may be the reason for lower breakdown fields. Such phenomena have not been previously reported by any other authors in the polymers under consideration.

Another direction of research that is useful to consider when using the polymers in question to create memory elements is associated with the transition from polymers to oligomers. It is known that, during the synthesis of the above class of polymers, various kinds of defects appear in the chains, among which include “head-to-head” (HH) defects, as well as oxygen-containing groups [21]. It follows from general considerations that as the molecular chain length decreases, the probability of HH-type defects should decrease. With the proper selection of crystallization conditions, it is possible to eliminate multiple nucleation and thus reduce the proportion of the amorphous phase. The polarization characteristics of such films were considered in [15]. It was found that the loops have a high degree of rectangularity, and their spontaneous polarization has a value close to the theoretical value. This should be attributed to low crystallization temperatures (– 190 °C). In this case, all forms of mobility in the oligomer are frozen and the surface is obtained defect free. Similar hysteresis figures on such oligomer films were also obtained in another work [16]. High stability during multicycle polarization switching (up to 10^5^) is maintained in the temperature range of 20–60 °C [16].

All the cited works utilized thin films that were deposited on one or another substrate and, therefore, the crystallization conditions of both sides of the film were different and often could not be controlled structurally. In the present work, the goal was to study the mechanisms of the electrical aging of block films of copolymers of vinylidene fluoride with tetrafluoroethylene (VDF-TFE) and vinylidene fluoride with hexafluoropropylene (VDF-HFP) copolymers with thicknesses of 15–50 μm. The emphasis was placed on the elucidation of the molecular mechanisms of structure formation on different sides of the film, which was monitored by attenuated total reflectance (ATR) spectroscopy. These results were supplemented with surface potential data obtained by the KPFM.

This work also considers the hypothesis of the possibility of asymmetric polarization switching. The considered polymers belong to the class of ferroelectrics, where the depolarization field E_dep_ is always present, and, therefore, the local field E_loc_ is written as Equation (1):E_loc_ = E_ext_ + E_dep_ + E_sc_,(1)
where E_ext_ is an external field and E_sc_ is space charge field formed by impurity carriers (catalyst residues, etc.), as well as at high fields by injected charges. Since polymers (even crystallizing ones) belong to low-dimensional systems, the representation of energy relations at the metal–polymer boundary in terms of the zone theory is not applicable here. It is customary to operate in terms of the highest occupied molecular orbital level in the valence band of the polymer (HOMO) and the lowest not occupied level in the conduction band (LUMO) [22]. On the boundary of the metal–polymer contact, the levels of the molecular orbitals of the polymer bend. The structure of the energy levels in this boundary can be changed by the injection phenomenon at sufficiently high fields. If both sides of the film differ in terms of their microstructure, then the injection phenomenon differs too. At the same electrode material and symmetric external field E_ext_, the carrier injection efficiency on both sides of the film will be different. Accordingly, asymmetric polarization switching becomes possible.

## 2. Materials and Methods

In this work, films of two copolymers, VDF-TFE and VDF-HFP, were studied. The films of the VDF-TFE copolymer were produced by extrusion. A standard extruder with a slit head was used. To study the effect of the form factor, two different film thicknesses were investigated: 16 microns and 58 microns. Samples of the VDF-HFP copolymer were obtained by watering in Petri dishes. The films were prepared by crystallization from a solution at room temperature by slow removal of the solvent. The initial concentration of the polymer was 4% by weight. Annealing was not carried out separately. The films were obtained from two different solvents—tetrahydrofuran (THF) and methyl ethyl ketone (MEK). The thickness of these films was about 30 μm.

The films of copolymers of vinylidene fluoride and tetrafluoroethylene, the microstructure of which was previously investigated by ^19^F NMR, where the VDF-TFE component ratio was 94-6, served as the objects of study [23]. The copolymer of VDF with hexafluoropropylene of the composition VDF-HFP 92-8 was also investigated [24].

X-ray diffraction measurements (XRD) were performed on a Bruker D8 Advance equipped with automatic slits (λ = 0.1542 nm) and position-sensitive LynxEye detector (with aperture angle of 3°). Reflection geometry was used. The scanning range was from 7 to 60°. The crystallite size *l*_hkl_ in the direction normal to the hkl plane was determined using the Debye–Scherer equation:(2)$$l_{c}=\frac{0.9k\lambda }{\cos\theta \sqrt{\beta^{2}-\beta_{et}^{2}}},$$
where *k* is the diffraction order, *λ* is the wavelength, and *β* and *β_et_* are the measured and etalon peak full widths at half maximum.

ATR spectra were obtained on a VERTEX 70 (Bruker Corporation, Karlsruhe, Germany) IR Fourier spectrometer in the range 4000–400 cm^−1^, using a PikeGladyATR single reflection attachment (PIKE Technologies, Fitchburg, WI, USA) with a diamond working element. The spectra were corrected using the program included in the OPUS 7.0 software to take into account the dependence of the penetration depth of radiation into the sample on the wavelength. A surface polymer layer (0.5–2 μm thickness) was scanned. Absorption spectra were measured on a TENZOR 37 IR Fourier spectrometer (Bruker Corporation, Karlsruhe, Germany).

Topography and KPFM mappings of the polymer sample were carried out with a scanning nanolaboratory Ntegra Prima (NT-MDT SI, Zelenograd, Russia) using an NSG10/Pt (Tipsnano, Tallinn, Estonia) platinum conductive probe with a spring constant of 12 N/m. For KPFM measurements, the probe scanned the surface topography using the tapping mode first and then a 1 V AC voltage was applied on the probe near its resonance frequency (~180 kHz) to measure the sample’s surface potential distribution through a DC voltage feedback loop. The scan rate was set to 0.5 Hz, and a lift scan height of approximately 50 nm was adopted. The topography and KPFM images were processed using the Gwyddion 2.63 software.

Typically, multi-cycle switching measurements are used to evaluate polarization switching stability. In the presented case, when a bipolar field in the form of rectangular pulses was applied to the film, the change in the switching curve was monitored as the amplitude of the external field increased. The modernized Sawyer–Tower scheme was used.

## 3. Results and Discussion

The tasks were set to follow the change in the high-voltage polarization characteristics during film crystallization in the polar β-phase. Such films were obtained from a VDF-TFE copolymer with different thicknesses: 16 and 58 µm. The IR spectroscopy data for the thinner film are presented in Figure 1, and show that crystallization was in the polar β-phase. This was evidenced by the presence of bands characteristic of the planar zigzag conformation: 442, 470, 510 cm^−1^, and 840, 1275, and 1400 cm^−1^ [1,2,3,4,5] (not shown in the figure). Along with the marked bands, the spectrum also showed bands present at 410 and 490 cm^−1^ that characterized the chains in the TGTG^−^ conformation. Also, the bands at 813 and 1235 cm^−1^ (not shown in the figure) that characterize the T_3_GT_3_G^−^ conformation are visible [1,2,3]. They are attributed to the presence of chains in the amorphous phase, as judged by the 600 cm^−1^ band, or in the metastable paraelectric phase (discussed below). A comparison of the ATR data on both sides of the film (Figure 1b) shows that the surfaces are not identical in terms of the chain microstructure (compare the absorbance in the 470 and 490 cm^−1^ region).

The spectroscopic data of a thicker film of the copolymer under consideration suggest a similar tendency. Taking this into account, it makes sense to compare the surface potential distribution patterns (and their histograms) on both sides of the films under consideration. They are shown in Figure 2 and Figure 3. The y-axis in the presented histograms shows the number of points (pixels) on the scan with a certain value of the surface potential. This intensity calculation is described in detail in [25,26].

Both films contain an asymmetric distribution of the surface potential depending on the selected side. Since the surface charge density (and, accordingly, the surface potential) will change the characteristics of the barrier at the polymer–metal interface, an asymmetric switching of the remanent polarization can be expected in accordance with Equation (1) when a bipolar field is applied to the sample. This can be seen for the thin film in Figure 4a at a field amplitude of 63 MV/m. Since, for the thicker film, the ratio of the surface potentials is qualitatively similar (Figure 2 and Figure 3), we should expect similar behavior for it as well. Figure 4b shows that such asymmetry does appear at a field of 120 MV/m. It is accompanied by an irreversible increase in the negative potential, which is attributed to the occurrence of pre-breakdown phenomena. Previously, we observed a similar behavior on extruded PE films.

The asymmetry of polarization switching with increasing negative potential observed in the thick film of the VDF-TFE copolymer at a field of 120 MV/m should not depend on the shape of the signal of the external field applied to the film according to general considerations. The hysteresis characteristics are most often studied at bipolar triangle voltages.

A detailed analysis of the shape of the D-E curve at fields higher than the coercive field according to different authors has shown the same pattern: the hysteresis curve has a non-closed form, and each subsequent cycle is accompanied by an increase in the negative potential [20,27,28,29,30,31,32,33,34,35,36]. That is, the same regularity is observed as in Figure 4b. This makes it possible to consider that the fluorine-containing polymers under consideration have common patterns of high-voltage polarization. It is known that they are characterized by high coercive fields, often approaching breakdown fields. In this connection, the general pattern of electrical breakdown in polymer dielectrics in general should be considered [37,38,39,40,41]. With respect to the polymers under consideration, one of the reasons for the observed increase in the negative potential at high fields can be attributed to the transfer of holes from the electrode to the polymer surface [39,40,41].

The noted results were obtained on the VDF copolymer crystallized in the polar β-phase. As a rule, all the measurements are performed on such samples, which are used for the preparation of sensors and actuators [1,2,3,4,42]. However, it is of interest to see the dielectric response also in the case when crystallization takes place in the “non-electroactive” α-phase. The question here is not so unambiguous, since there are reports that hysteresis loops are also observed in this case (see, for example, the data of [31]). The results obtained on such films by KPFM also indicate the presence of hysteresis phenomena in them with sufficiently high values of the effective constant d_33_ [41]. The mechanism of the emerging piezo effect proposed in this work, associated with the movement of kink defects along the c-axis of the crystal, could be related to its longitudinal size.

In this regard, this part of the work followed the features of high-voltage polarization on two VDF-HFP copolymer films, which differed markedly in crystal size. This was achieved by its crystallization in different solvents: tetrahydrofuran and methyl ethyl ketone. The data on them are presented in Figure 5. From this, after the separation of the overlapping reflections into components by Equation (2), the average sizes of the crystals in the different directions were calculated, which are presented in Table 1. In the film crystallized from THF, these sizes are several times lower.

IR spectroscopy data (Figure 6), however, allow us to find the details of the microstructure of the chains in the amorphous phase as well. Since the crystallization of both films occurs in the α-phase, the 510, 813, and 840 cm^−1^ bands will characterize the presence of isomers with a T_3_GT_3_G^−^ and planar zigzag conformation exactly in the amorphous phase or metastable paraelectric phase. This is an important observation, since, when an external field is applied, the marked isomers also participate in the polarization process. As follows from Table 2, the concentration of the marked isomers appears to be higher in the amorphous phase of films crystallized from MEK.

Taking this into account, let us compare the high-voltage polarization curves in both films (Figure 7). It follows that, at the specified field, the charge response of the film crystallized from solution in THF reveals a tendency to accumulate negative potential, where the shape of the current response curve indicates the appearance of the microchannels of the breakdown. It can be assumed that the latter localize in the amorphous phase of the crystallizing copolymer with a reduced local density in its amorphous phase so that the mentioned microchannels of breakdown can occur. These are packing defects of a dynamic nature, but along with them in the considered polymers are topological packing defects too. This type of defect exists along the boundaries of the end surfaces of neighboring lamellar crystals and participate in the formation of the so-called “large period” [3]. Such regions with reduced (relative to the crystal) packing density can also initiate breakdown microchannels.

Comparing Figure 7 and Figure 8 makes it clear and illustrates that the film crystallized from MEK, at the beginning of the negative potential accumulation with weak current oscillations, is shifted to higher fields. On the other hand, it is shown in Figure 7b and Figure 8b that the conductivity in the films crystallized from THF is several times higher than that in the films crystallized from MEK. It can be assumed that the impurity carriers (and partially injected from the electrodes into the polymer) will participate in the process of lowering the depolarization field created by the crystals. In this case, according to Equation (1), the film crystallized from THF should have a decreased depolarization field because of its intrinsic carriers against the external field (E_ext_). If the response curves (D,j) are registered by the field function (E_ext_) from an external source, this will be perceived as a shift in the response kinetics to lower fields (Figure 7a,b). The conductivity of the film crystallized from MEK is several times lower for the film crystallized by THF (Figure 7b); in this case, the depolarization field due to intrinsic carriers will decrease weakly. The indicated conduction anomaly in the case of MEK will be observed at higher fields (Figure 8a,b).

From these data, it is possible to obtain information on the mechanisms of conductivity in the films under consideration. The emphasis is placed on revealing the role of defects in the chemical attachment of the neighboring links in the head-to-head-type defects (HH). In earlier work [23,24], the estimation of their fraction by the intensity ratio of the 2920 and 3024 cm^−1^ valence vibration bands of methylene groups was proposed. Here, the latter is characteristic of the CH_2_ groups in the “normal” position, while the 2920 cm^−1^ band is associated with vibrations in the defective (-CH_2_-CH_2_-) attachments [4,42,43] and which was observed, for example, in PE [17]. Using the absorption data, it is possible to obtain information on the relative change in the marked defects in the average volume. Figure 6a and Table 2 show that the film crystallized from THF has more of these defects than the film crystallized from MEK. It is shown in Table 1 (the last column) that the proportion of the amorphous phase (metastable paraelectric phase) for a film crystallized from THF is higher. Considering that, among the attachment defects, there will be -CH_3_ groups, which, for steric reasons, will prevent the dense packing of chains of the amorphous phase, the THF film should have an increased free volume. If considering that the fraction of such a phase in this film is higher (Table 1), the impurity conductivity due to drift mobility in it will be higher than that in the film crystallized from MEK under the same external field conditions. The large size of the crystals in the latter (Table 1), which play the role of traps for carriers, also contributes to the decrease in conductivity in it. In the whole field range, the conductivity in the film obtained from MEK is lower than that crystallized from THF. As a result, the Joule heat in the amorphous phase (metastable paraelectric phase) regions of the latter film will induce the appearance of pre-breakdown microchannels at the lower fields, which is consistent with our experiment (Figure 7 and Figure 8).

The role of the packing density of the amorphous phase in these polymers is also confirmed by comparing the X-ray data for the VDF-HFP and VDF-TFE copolymers (Figure 5a). It is shown in these data that the packing density of the chains of the amorphous or metastable paraelectric phase in the VDF-TFE copolymer is higher than that of the VDF-HFP copolymer. This is evidenced by the fact that the angular position of the halo in the VDF-HFP copolymer is shifted to lower values by almost 2 degrees than that for the film crystallized from THF or MEK. In the context of the above, the conductivity in the VDF-TFE copolymer, other things being equal, should be lower and the appearance of pre-breakdown microchannels in it will be observed at higher fields, which agrees with our experiment (Figure 4b). It should be added that the noted pre-breakdown microchannels can be registered also by the characteristics of electrical noise [44]. The appearance of noise during polarization switching is also observed in inorganic ferroelectrics [45,46]. Such phenomena are called the Barkhausen effect; they are associated with the movement of the domain wall of the ferroelectric domain. Unlike inorganic materials, similar phenomena were only observed in the polymer films when a negative voltage was applied. This effect is interesting and requires further study.

The polymer chains located in the film surface can participate in the mentioned processes, since, at sufficiently high fields, the carriers injected from the electrode material will be added to their own carriers. In this connection, the surface microstructure of the films was analyzed using ATR IR spectroscopy. As before, the emergence of head-to-head defects on the film surface was monitored.

If the noted defects extend to the electrode surface, there is a possibility of proton transfer to the electrode.

Figure 9 shows the comparative ATR curves (averaged on both sides) for the films crystallized from MEK and THF. The ratio of the intensities of the 2920/3024 bands is higher in the latter case. This means that the interface of the THF-derived film contains more defect protons. In this case, there is a higher probability of their transfer to the electrode. The resulting current will contribute to the reduction in the breakdown voltage.

## 4. Conclusions

The observed effect of electrical aging on ferroelectric polymers was studied in films of two different VDF copolymers, with varying degrees of crystallinity, ratios of ferroelectric and paraelectric phases, and numbers of defects. This choice is based on the difference in the structure of the materials used and the ease of understanding of the impact of the specified factors in film production on its structure.

Asymmetry in polarization switching is observed for all the samples. The kinetic curves of this and the charge response differ markedly when applying a potential of different signs. When a negative potential is applied, pre-breakdown phenomena similar to Barkhausen noises are observed.

The copolymer films obtained from the THF and MEK solutions, according to the X-ray and IR spectroscopy data, crystallize in a nonpolar α-phase. However, polarization switching is also observed in these films. For the films from different solvents, the ratio of various rotatable isomers in the amorphous phase varies. This circumstance affects several characteristics of arbitrary polarization. In particular, the noted pre-breakdown phenomena are observed at lower fields in the case where the film has crystals of a small size.

This paper has shown that the value of the breakdown field of the polymer is influenced by the conditions of film production, such as the type of solvent, method, and the thickness of the film. Based on the data obtained, it can be concluded that to produce ferroelectric memory devices, it is preferable to use films made from the VDF-TFE copolymer, as they crystallize in a polar β-phase. This conclusion is supported by the data (Figure 4), which show that pre-breakdown phenomena occur at higher electric fields for this material.

## Figures and Tables

**Figure 1 nanomaterials-14-01002-f001:**
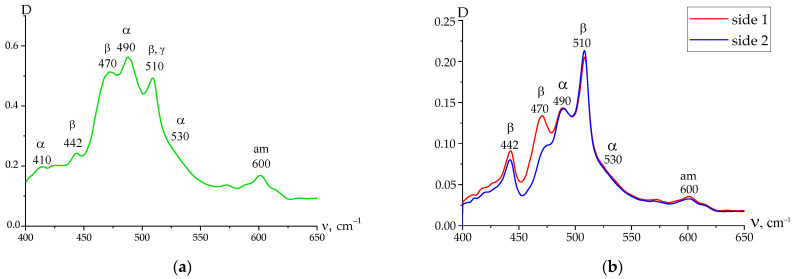
IR spectra of VDF-TFE copolymer thin film: (**a**) absorption spectrum; (**b**) comparison of ATR spectra of the 1st and the 2nd sides of the 16 μm thick film.

**Figure 2 nanomaterials-14-01002-f002:**
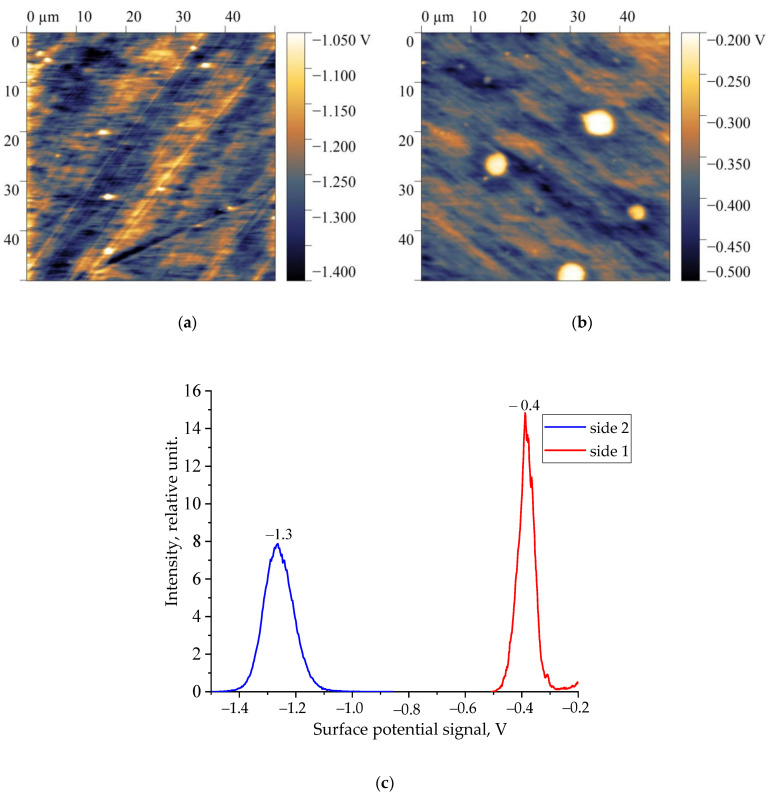
Surface potential distribution maps: (**a**) side 1, (**b**) side 2, and (**c**) histograms for 16 µm thick film of VDF-TFE copolymer.

**Figure 3 nanomaterials-14-01002-f003:**
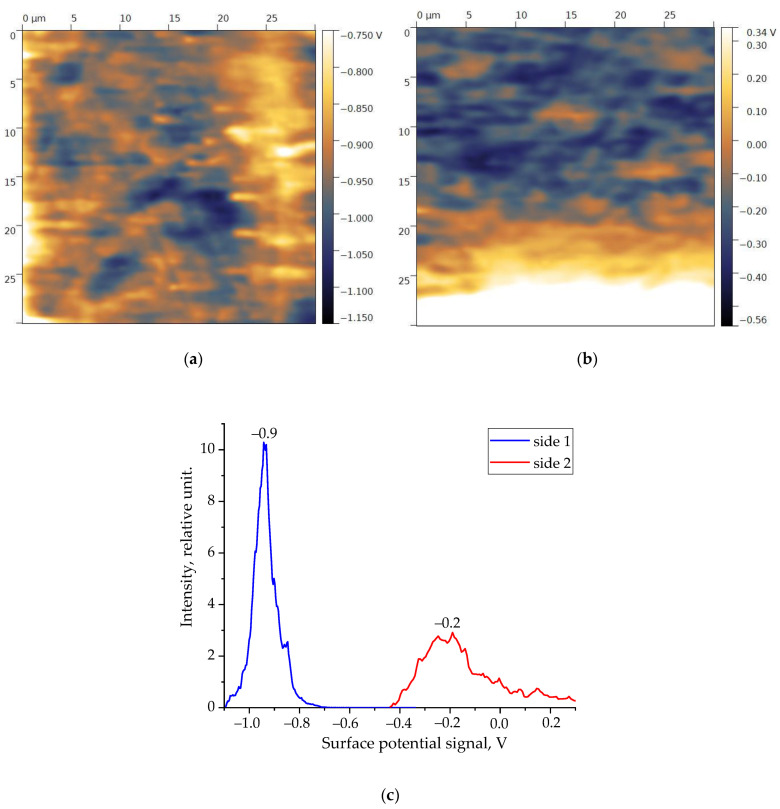
Surface potential distribution maps: (**a**) side 1, (**b**) side 2, and (**c**) histograms for 58 µm thick film of VDF-TFE copolymer.

**Figure 4 nanomaterials-14-01002-f004:**
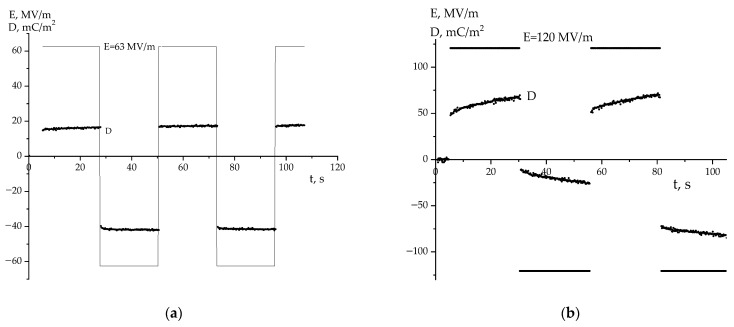
Kinetic curves of surface charge density variation in a 16 (**a**) and 58 (**b**) µm thick VDF-TFE copolymer film.

**Figure 5 nanomaterials-14-01002-f005:**
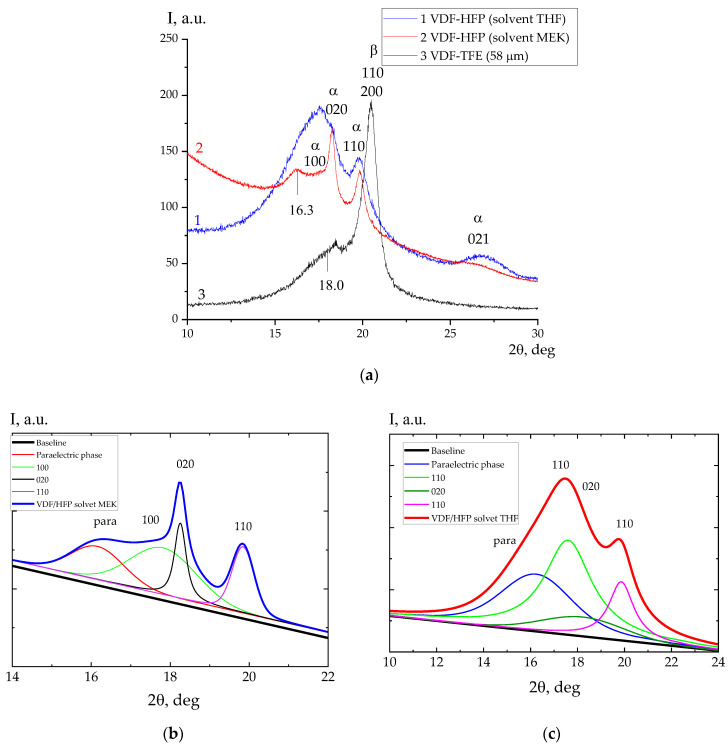
(**a**) X-ray diffraction from VDF/HFP copolymer films obtained by crystallization from solution in THF (1), MEK (2); curve 3 from VDF-TFE copolymer film. (**b**,**c**) are diffraction curves of films obtained from MEK (**b**) and THF (**c**).

**Figure 6 nanomaterials-14-01002-f006:**
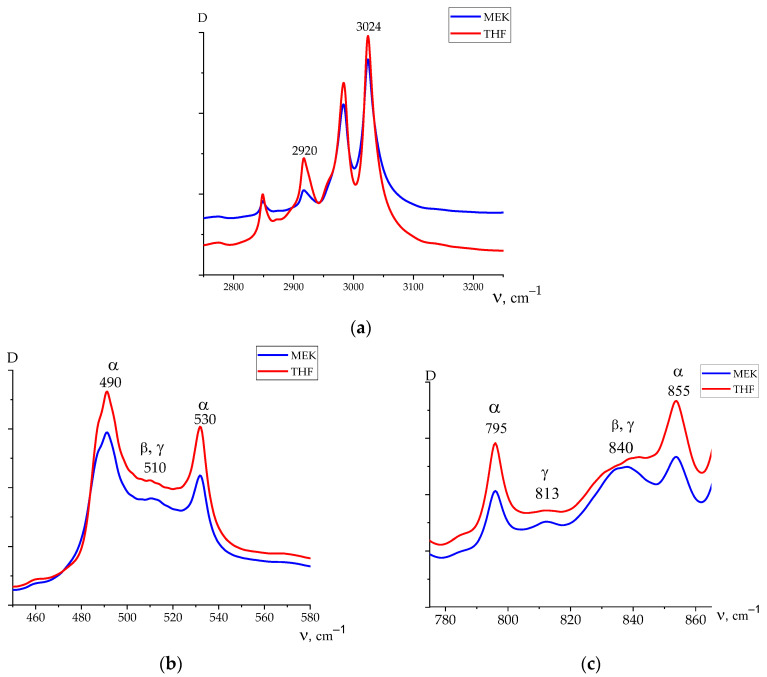
Absorption spectra in different spectral regions (**a**,**b**,**c**) for films crystallized from THF and MEK.

**Figure 7 nanomaterials-14-01002-f007:**
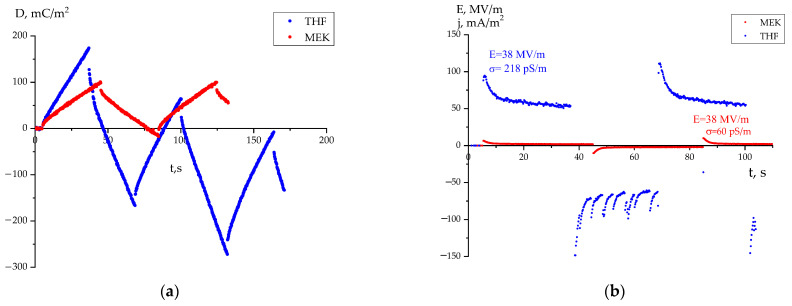
Comparison of high-voltage polarization kinetics from surface charge density (**a**) and current (**b**) measurements at 38 MV/m rectangular field pulses applied to VDF/HFP copolymer films prepared from different solvents.

**Figure 8 nanomaterials-14-01002-f008:**
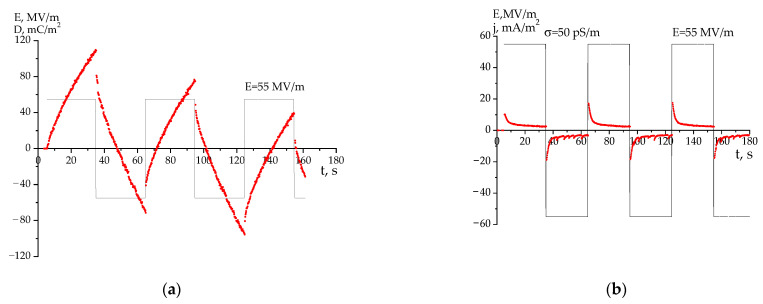
Comparison of kinetics of high-voltage polarization charge (**a**) and current (**b**) response in MEK-derived VDF-HFP copolymer films to an external field of 55 MV/m.

**Figure 9 nanomaterials-14-01002-f009:**
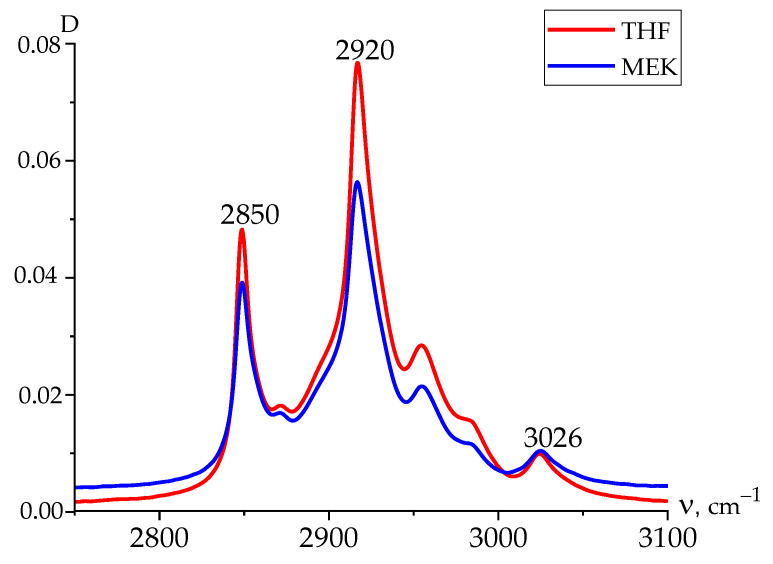
ATR IR spectra of VDF-HFP copolymer film crystallized from solution in different solvents averaged on both sides.

**Table 1 nanomaterials-14-01002-t001:** Structural characteristics of VDF-HFP copolymer films of composition 92-8 obtained from different solvents.

Solvent	l_hkl_, nm			l_p_, nm	χ_p_
	l_100_	l_020_	l_110_		
MEK	4.1	23.0	12.2	5.1	0.23
THF	3.3	1.9	6.3	2.2	0.32

**Table 2 nanomaterials-14-01002-t002:** Ratios of intensities of conformationally sensitive absorption bands in VDF/HFP copolymer films obtained by crystallization from different solvents.

Solvent	D510β,γD530α	D813γD795α	D840β,γD855α	D2920D3024
MEK	0.81	0.25	1.0	7.10
THF	0.65	0.1	0.5	9.42

## Data Availability

The data that support the findings of this study are available from the corresponding authors upon reasonable request.

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
