# Peer review of "The Effect of Electric Aging on Vinylidene Fluoride Copolymers for Ferroelectric Memory"

_nanomaterials, 2024, doi:10.3390/nano14121002_

Round 1
Reviewer 1 Report
Comments and Suggestions for Authors
The manuscript submitted by Kochervinskii et al reported the effect of the structure of self-polarized films of copolymers of vinylidene fluoride with tetrafluoroethylene and hexafluoropropylene on breakdown processes. Results show that the value of the breakdown field of the polymer is influenced by the conditions of film production. Such as type of solvent, method, thickness of film. The manuscript was well organized but still have some problem. Suggestions and comments are listed as below:
1.Too many self-citations in the reference list.
2.Since that there is a phase “ferroelectric memory” in the title, the ‘ferroelectric’ and ‘memory’ properties should be investigated.
3.What is the “intensity” refers to in the y-axis of figure 2 and figure 3.
4.The thickness of the as-prepared thin film should be checked.
Author Response
The authors thank the reviewer for his attention to this work.
1. We agree with the remark. The list of references has been adjusted.
2. The submitted work focuses on the study of ferroelectric polymer materials that could be used as functional materials for ferroelectric memories. The development of such memories is a promising research area, and there are already money studies in this field. These studies emphasize the number of switching cycles in materials used for ferroelectric memory elements. In the proposed manuscript, the authors emphasize the influence of external fields on these processes. Ferroelectric polymers, similar to other ferroelectric materials, have two directions of spontaneous polarization, which allow them to transition between states under the influence of an electric field. This process is like how magnetic materials operate with magnetic moments, which are used in ferromagnetic memory devices. The proposed polymer materials can replace the currently used ferromagnetics, and devices based on them can show great performance. Therefore, we consider it important to study their properties.
3. The «intensity» indicated by the y-axis corresponds to the number of points or pixels on the scan with a certain value of the surface potential. This is described in more detail in DOI: 10.1016/j.j mat.2016.08.001 and DOI: 10.1039/c6nr09310h. This remark has been added to the text.
Reviewer 2 Report
Comments and Suggestions for Authors
This work focuses on studying the stability and aging effects of different copolymer films, including their performance under various conditions and potential applications in ferroelectric memory devices. Further exploration of these areas can provide valuable insights for future research and applications.
1. I suggest exploring the electrical aging effects of copolymer films under varying frequencies or voltage conditions. Please determine if there are consistent performances or differences under these conditions.
2. It is recommended to analyze the specific characteristics and mechanisms of pre-breakdown phenomena.
3. I recommend performing a comparative analysis on the electrical aging effects of copolymer films under different environmental conditions. Identify any significant performance differences under various temperature, humidity, or atmosphere conditions.
4. Please consider the renewability and cycling performance of copolymer films. Conduct multiple cycling experiments to evaluate their stability and durability.
5. I recommend to consider the influence of impurities in copolymer films on their electrical performance and aging effects. Conduct relevant analyses and control experiments.
6. This work investigated the ferroelectric nanocomposites enabled functional devices. Some relative papers may enrich the concepts and background of this work as references: Nano Energy, 2023, 116: 108788.
Comments on the Quality of English Language
Some grammar and typing errors should be modified.
Author Response
The authors thank the reviewer for the careful analysis of the manuscript. The grammar and typing errors have been corrected.
1. The question is correct. In this paper, the task was to compare the appearance of "anomalies" in the processes of switching spontaneous polarization with the structural features of the films under consideration. The frequencies, as can be seen from the work, were 10-100 ms. In principle, such studies can be carried out at higher frequencies. This requires other equipment. Such work is planned in the future and we are looking for partners for such joint research.
2. The remark is correct and at that moment we are starting a cycle of research on the study of the mechanisms of electrical breakdown in this materials.
3-5. Thank you so much for your comments. At that moment, we have begun to carry out work on the influence of the quality (purity) of the solvent from which films are made on their electrical and structural characteristics. Preliminary data show that this is an important and extensive field of activity. Unfortunately, we cannot provide this data yet. But as soon as there are sufficient results in this area, they can become material for a new study.
6. Thank you for sending the link. We will definitely use its results. The article concerns sensors based on PVDF-PZT composite. At that moment, we are also planning to work to composite materials based on VDF copolymers and piezoceramics. The work is in the initial stage and there are no results yet.
Reviewer 3 Report
Comments and Suggestions for Authors
Comments
The paper deals with ‘The effect of electric aging on vinylidene fluoride copolymers for ferroelectric memory’, by Valentin V. Kochervinskii, Evgeniya L. Buryanskaya, Aleksey S. Osipkov, Mstislav O. Makeev, Dmitry A. Kiselev, Margarita A. Gradova, Oleg V. Gradov, Boris V. Lokshin, Alexandr A. Korlyukov
1. Introduction
P 2/16, Line 64. ‘while PVDF can 64 be formed in various polymorphic modifications [1,2].’
PVDF can have various forms. Can the authors speak about these forms, and the needed mechano-thermal conditions needed to have them?
P 2/16, Line 66. ‘of an agent providing 66 the formation of a polar modification [1,2].’
Can the authors speak about this/these agents/s.
Only two references cited, maybe the works are larger.
P 2/16, Line 93. ‘…should be attributed to low crystallization temperatures (– 190С).’
The very low temperature, -190C, how to measure it? For what application/interest can we find such a low temperature?
P 3/16, Line 98. ‘…All cited works were performed on thin films…’
What is the films thickness?
The thickness has an important effect? On what?
2. Materials and Methods
P3/16, Line 126. ‘films of the VDF-TFE copolymer were produced by extrusion’
What are the process parameters versus film thickness? Extrusion machine name?
P3/16, Line 128. ‘…Samples of the VDF-HFP… two different solvents – tetrahydrofuran (THG) and methyl ethyl ke-129 tone (MEK)’
Can the authors explain how they did? Concentration of dissolution/annealing/temperature… Both of the solvents were used? Only one? Which one seems better, etc.
How was done the crystallization?
P3/16, Line 133. General comments about ‘The copolymer of VDF with hexafluoropropylene of the composition VDF-HFP 92-8 was also investigated’
The composition is important, isn’t it? How the find the best composition for the best required properties? How to control this composition and make it?
3. Results and Discussion
P4/16, Line 179. Figure 1. ‘the 1st and the 2nd sides of the film thick 16 μm.’
It is the film from extrusion? Side 1 and side 2 have same condition of process in extrusion, the reviewer means if any contact with a metallic support and/or or air etc. during extrusion…
P5/16, Line 181. ‘The spectroscopic data of a thicker film of the copolymer under consideration give a similar tendency [26].’
Can the authors describe a little this refence [26], please? To help the further Figures 2 and 3 discussion.
P5/16, Line 186. Figure 2. ‘Surface potential distribution maps (a) - side 1, (b) - side 2 and (c) - histograms for 16 μm thick film of VDF-TFE copolymer.’
And
P6/16, Line 188. Figure 3. ‘Surface potential distribution maps (a) - side 1, (b) - side 2 and (c) - histograms for 58 μm thick film of VDF-TFE copolymer.’
Can the authors remember us the way to make the observations and measurements.
The observations are a little blurred.
P7/16, Line 206. Figure 4. ‘Figure 4. Kinetic curves of surface charge density variation in a 16 (a) and 58 (b) μm thick VDF-TFE 206 copolymer film.’
Can the authors add for 16µm data for E=120 mV/m?
Can the author add for 58µm data for E=63 mV/m?
Especially with (16µm data for E=120 mV/m) maybe is it possible to emphasize the comments that:
Line 213. ‘the fluorine-containing polymers under consideration have common regularities of high-voltage polarization’ /
Line 218. ‘increase in the negative potential at high fields can be attributed to the transfer of holes from the polymer surface to the electrode [41-43].’
P10/16, Line 270 and Line 273. Why is used 38 and 55 mV/m?
Why not 63 and 120 mV/m?
What is the film thickness?
P11/16, discussion is based on references. Maybe it is important to check the materials states (phase, process, thickness, etc.), isn’t it?
For comments/discussion: The information can be added by the authors? Or the authors think it is not so important, and why if it is.
What is original in the present paper? VDF-HFP?
4. Conclusion
Maybe it can be improved with key results materials & data, and better link with states of the materials: solvent method>>how to fox materials crystallization and thickness, how to fix analysis and compare same analysis conditions, if no why.
P12/16 Line 361. ‘Paper is shown that the value of the breakdown field of the polymer is influenced by the conditions of film production.’
Author Response
The authors thank the reviewer for the careful analysis of the manuscript.
Introduction
1. It is a good question. In this paper, the task was to compare the appearance of "anomalies" in the processes of switching spontaneous polarization with the structural features of the films under consideration. The frequencies, as can be seen from the work, were 10-100 ms. In principle, such studies can be carried out at higher frequencies. This requires other equipment. Such work is planned in the future and we are looking for partners for such joint research.
2. Here we are talking about some additives that contribute to the crystallization of PVDF in the polar β-phase in the isotropic state. Indeed, there are more such works, but we took links to the works that were published in the form of books.
3. The film crystallized from a melt at a low temperature, and hysteresis measurements were carried out at a higher temperature.
4. The thickness of the film can affect the measured parameters. For example, the breakdown voltage is significantly lower for a thinner film. However, we noted in our article that the crystallization conditions on both sides of the film were different. The structure of the side facing the substrate was not controlled, so we worked with free-standing films, where it was possible to evaluate the structure from both sides.
Materials and Methods
5. It was a standard extruder with a slit head.
6. The films were prepared by crystallization from a solution at room temperature by slow removal of the solvent. The initial concentration of the polymer was 4% by weight. Annealing was not carried out separately. This information was added in text
7. Data on the composition of the copolymer were obtained earlier in another article by the 19F NMR method. This date present in https://doi.org/10.3390/polym16020233 and https://doi.org/10.3390/nano13212851
Results and Discussion
8. We proceeded from the fact that the structure of the film on both sides differs and did not study which cooking processes are responsible for this.
9. In both films (of different thicknesses), bands responsible for the presence of conformations TGTG-(410, 490 cm-1), T3GT3G- (430, 813 and 1235 cm-1) and a planar zigzag (440, 840 and 1275 cm-1) were always present in the ATR spectra.
10. Studies of the surface potential were carried out using the Kelvin probe method. This technique is described in detail in the works 10.1016/j.j mat.2016.08.001 and DOI: 10.1039/c6nr09310h
11. We can add date for 58 µm data for E=63 MV/m, but this curve is typical. We think that such data do not describe pre-breakdown phenomena. Such curve is unnecessary. Unfortunately, we cannot give these curves, since a breakdown occurred for a thinner film at fields of 120 MV/m.
12. A careful analysis of the hysteresis curves for a large number of cited papers shows that the curves have an unclosed form, and each subsequent cycle is accompanied by an increase in negative potential. This fact can be qualitatively explained by the phenomenon of injection of hole carriers [41-43].
13. These fields are chosen because at such values, a decrease in the surface potential with negative polarity is visually visible. The thicknesses of the films are indicated in the methodological part of the work. The films thickness were about 30 microns.
14. It is shown that the electrophysical characteristics of ferroelectric films are controlled by the structure, including in their surface
15. In this article, an original approach is presented in which subtle variations in the microstructure of the film's surface are believed to be responsible for changes in its high-voltage polarization and electrical conductivity. The example of VDF-HFP copolymer films shown that the size of crystallites can significantly affect the critical fields at which pre-breakdown phenomena begin to manifest themselves.
Conclusion
16. Using the example of the materials under consideration, it was possible to show what subtle differences in the microstructure of the surface of the formed films are responsible for the change in high-voltage polarization and conductivity.
Round 2
Reviewer 1 Report
Comments and Suggestions for Authors
Since that all of my previous concerns have been settled in the revised manuscript, I agree this article to be published on Nanomaterials.
Author Response
Thank you so much for your work with our the manuscript!
Reviewer 3 Report
Comments and Suggestions for Authors
Thanks. The authors reply to questions.
Be careful : page 11/16 line 299 into [24.25 ????]
Author Response
Thank you so much for your work with our the manuscript!
The page 11/16 line 299 into [24.25] was checked.